# Determinants of COVID-19 Vaccine Hesitancy: A Cross-Sectional Study in Three Communities in the United States and Lebanon

**DOI:** 10.3390/microorganisms12061200

**Published:** 2024-06-14

**Authors:** Mohamad Yasmin, Mohamad Ali Tfaily, Rayyan Wazzi Mkahal, Rita Obeid, Rebecca P. Emery, Habiba Hassouna, Mudita Bhugra, Robert A. Bonomo, Zeina A. Kanafani

**Affiliations:** 1Louis Stokes Cleveland VA Medical Center, Cleveland, OH 44106, USA; mxy312@case.edu (M.Y.); rab14@case.edu (R.A.B.); 2Department of Internal Medicine, Emory University, Atlanta, GA 30322, USA; mohamad.ali.tfaily@emory.edu; 3Faculty of Medicine and the Medical Center, American University of Beirut, Cairo Street, Riad El Solh, Beirut 1107 2020, Lebanon; rayyanmkahal@gmail.com; 4Department of Internal Medicine, Case Western Reserve University, Cleveland, OH 44106, USA; rita.obeid@case.edu (R.O.); mudita.bhugra@corewellhealth.org (M.B.); 5Spectrum Health, Michigan State University College of Human Medicine, Grand Rapids, MI 49503, USA; rebecca.emery@corewellhealth.org (R.P.E.); habiba.hassouna@spectrumhealth.org (H.H.)

**Keywords:** COVID-19, vaccination, vaccine hesitancy, survey, pandemic

## Abstract

The COVID-19 pandemic underscores the significance of vaccine hesitancy in shaping vaccination outcomes. Understanding the factors underpinning COVID-19 vaccination hesitancy is crucial for tailoring effective vaccination strategies. This cross-sectional study, conducted in three communities across the United States and Lebanon, employed surveys to assess respondents’ knowledge, attitudes, and perceptions regarding COVID-19 infection and vaccination. Among the 7196 participants, comprising 6775 from the US and 422 from Lebanon, vaccine hesitancy rates were comparable at 12.2% and 12.8%, respectively. Notably, a substantial proportion of respondents harbored misconceptions, such as attributing the potential to alter DNA (86.4%) or track individuals (92.8%) to COVID-19 vaccines and believing in the virus’s artificial origins (81%). US participants had more misconceptions about the COVID-19 vaccine, such as altering DNA or causing infertility. Lebanese participants were more likely to question the origins of the virus and the speed of vaccine development. Additionally, US respondents were less worried about infection, while Lebanese respondents were more indecisive but less likely to outright reject the vaccine. Primary determinants of hesitancy included perceptions that the vaccine poses a greater risk than the infection itself (aOR = 8.7 and 9.4, respectively) and negative recommendations from healthcare providers (aOR = 6.5 and 5.4, respectively). Conversely, positive endorsements from healthcare providers were associated with reduced hesitancy (aOR = 0.02 and 0.4, respectively). Targeting healthcare providers to dispel misinformation and elucidate COVID-19 vaccine risks holds promise for enhancing vaccination uptake.

## 1. Introduction

Vaccination has been extremely important to control COVID-19 infection since immunization is one of the most effective and affordable health strategies to prevent infectious diseases. Despite the record time in which the various COVID-19 vaccines were developed and the clinical trials were conducted [1,2], the vaccine rollout was not always as swift. Efforts to achieve widespread vaccination were stalled by several social, economic, and political factors [3,4,5], and one of the most notable obstacles was reluctance to receive the vaccine [6]. Several studies have looked at COVID-19 vaccine acceptance rates and have found that the numbers vary greatly by geographical region and population type [7]. In a global survey [8], the percentage of adults willing to receive the COVID-19 vaccine was higher in low- and middle-income countries (average of 80.3%) compared to the United States (64.6%) and Russia (30.4%). Among low- and middle-income countries, the lowest acceptance rates were observed in Burkina Faso and Pakistan (66.5%), while the highest was in Nepal (96.6%) and Mozambique (89.1%).

A number of studies have looked at the factors contributing to vaccine acceptance [9,10,11]. Higher perceptions of COVID-19 severity, increased likelihood of being infected, positive perceptions of protective behaviors against COVID-19 in the community, and self-engagement in protective behaviors against COVID-19, are all factors that influenced vaccination receptivity positively [12]. Married status, male sex, receiving the seasonal influenza vaccination, and valuing a physician’s recommendations were additional factors contributing positively to vaccine acceptance [13]. Race and a rising perception of the vaccine’s risks were factors that made people less eager to be immunized [12].

While reports on vaccine uptake from the United States are abundant, data from Lebanon are much scarcer. To our knowledge, there are no studies that examine the awareness, perceptions, and attitudes towards the COVID-19 vaccines among adults in Lebanon. The aim of this study is to compare the determinants of COVID-19 vaccination in the United States and Lebanon. Comparing these two countries provides valuable insights due to their contrasting socio-economic, cultural, and healthcare contexts, which can highlight universal versus region-specific determinants of vaccine hesitancy. Identifying factors associated with vaccination hesitancy will help in addressing the challenges of vaccine uptake and will inform vaccination campaigns at the local and international levels.

## 2. Materials and Methods

Study design and setting: We conducted a cross-sectional survey between March 2021 and June 2021 in three major urban areas in the United States and Lebanon served by university medical centers: Case Western Reserve University in Cleveland, OH, Spectrum Health at Michigan State University College of Human Medicine in Grand Rapids, MI, and the American University of Beirut Medical Center in Beirut, Lebanon. In the United States, the survey was disseminated electronically via ResearchMatch.org, a disease-neutral, web-based recruitment registry, while in Lebanon, surveys were sent using LimeSurvey. The catchment population included members of the general public, healthcare workers, and university students.

Survey: The survey included socio-demographic data, health-related characteristics, knowledge questions about COVID-19 infection and vaccine, as well as perceptions and attitudes towards COVID-19 infection and vaccine. To assess COVID-19 knowledge, a 10-item True–False questionnaire was included in the survey. Questions were prepared based on publicly available information available on the Centers for Disease Control and Prevention (CDC) and the World Health Organization (WHO) websites. A knowledge score was calculated for each respondent based on the number of questions answered correctly. For the primary endpoint variable pertaining to vaccination intent, respondents to the question “I am willing to take the COVID-19 vaccine” could select only one of four possible options: (1) definitely yes; (2) undecided/likely; (3) undecided/unlikely; (4) definitely no. Vaccine hesitancy was defined as any answer other than “definitely yes”.

Statistical analysis: Survey data analysis was performed using IBM SPSS Statistics 25 (SPSS Inc., Chicago, IL, USA). For categorical variables, the Chi-square test was used to compare the baseline characteristics, knowledge, perceptions, and attitudes in the US and Lebanese cohorts. For the continuous variable of knowledge score, the independent samples *t*-test was used to compare the means in the two groups. In each cohort, multivariable backward logistic regression was used to calculate adjusted odds ratios (aOR) and their 95% confidence intervals (CI) in the evaluation of independent associations between the different variables and vaccine hesitancy. Variables were included in the logistic regression model if the *p*-value on bivariable analysis was ≤0.20.3.

## 3. Results

A total of 7197 respondents responded to the survey, 6775 in the United States and 422 in Lebanon. The study population was equally distributed across all three strata for age (18–39 years, 40–59 years, and ≥60 years). Individuals identifying as female predominated with 75.7% of the study population, and healthcare workers comprised 26.8% of all survey respondents, while individuals with at least a college education represented 83.6% of the survey sample. Very few survey participants had a previous personal history of COVID-19 infection (9.0%), but a large proportion had a history of COVID-19 infection in family members or friends (65.0%). Overall, the vaccine acceptance rate was high (87.8%), with 4.4% rejection, and 7.8% indecisiveness, giving a vaccine hesitancy rate of 12.2%. The complete socio-demographic and health-related characteristics of the study population are shown in Table 1 classified by the study site.

Comparing the characteristics of the two cohorts, the Lebanese cohort was younger, less likely to consume alcohol but more likely to be smokers, and had fewer pre-existing medical conditions than the US cohort. As for the primary outcome of vaccine hesitancy, it was comparable between the two cohorts (87.2% vs. 87.8%), although Lebanese respondents were more likely to be indecisive (11.6% vs. 7.6%) than reject the vaccine (1.2% vs. 4.6%).

The vast majority of respondents were accurate in identifying the symptoms of COVID-19 infection and the mode of transmission (correct response rate above 90% for all questions). However, much fewer correct answers were recorded for questions pertaining to the safety of the vaccine, particularly in the US cohort, where a large proportion of respondents believed that the COVID-19 vaccine can change human DNA, can lead to infertility, or can be used to track individuals (range from 82.0% to 94.9% of the US cohort). Table 2 shows the knowledge about COVID-19 infection and vaccination in the study population.

As for the attitudes and perceptions about COVID-19 infection and vaccination (Table 3), most respondents agreed on the benefits and effectiveness of vaccination in general (95.8% and 94.6%, respectively). However, more respondents in the Lebanese cohort questioned the origin of SARS-CoV-2 (36.3% vs. 17.9%) and the development of the vaccine (51.4% vs. 37.8%). On the other hand, more US respondents perceived that their risk of becoming infected was low (61.3% vs. 24.4%). Importantly, the majority of individuals in both cohorts expressed their willingness to take the vaccine upon the recommendation of their healthcare provider (90.2% of the overall population).

In view of the significant differences in the baseline characteristics between the US and Lebanese cohorts, we elected to perform separate regression models for each cohort to determine the variables associated with vaccine hesitancy (Table 4). The variable with the strongest association with vaccine hesitancy in both cohorts was the perception that the COVID-19 vaccine was more dangerous than the COVID-19 infection, with an aOR of 8.7 among US respondents and 9.4 among Lebanese respondents. The other important determinant common to both cohorts was the advice from the healthcare provider not to take the vaccine, with an aOR of 6.5 among US respondents and 5.4 among Lebanese respondents. The remaining independent variables in the two cohorts were the belief that the vaccine development was rushed and being a healthcare worker. Interestingly, believing in the benefits of most routine vaccinations was significantly associated with vaccine hesitancy in the US cohort (aOR = 5.4). The knowledge score had a differential effect on hesitancy in the two subject groups, whereby there was a positive association in the US group and a negative association in the Lebanese group. However, the association was relatively weak in both groups (aOR 1.3 and 0.7, respectively). Variables that predicted less vaccine hesitancy, i.e., more vaccine acceptance, common to the US and Lebanese cohorts were a vaccine recommendation by a healthcare provider (aOR = 0.02 and 0.4, respectively), age (aOR = 0.6 and 0.4, respectively), and education (aOR = 0.8 and 0.6, respectively). Believing in vaccination as an effective strategy and receiving the influenza vaccine were negative predictors of vaccine hesitancy only in the US cohort (aOR = 0.3 and 0.6, respectively).

## 4. Discussion

Administering a cross-sectional survey to over 7000 individuals across three communities in two countries has allowed us to identify the major determinants of vaccination uptake and vaccination refusal during the peak months of the COVID-19 pandemic. During the timeframe of this study (April 2021–June 2021), a gradually rising proportion of the population in both countries had access to COVID-19 vaccines, yet a significant fraction seemed reluctant to receive the vaccine. Therefore, estimating vaccine hesitancy and its underpinnings is important in informing future vaccination campaigns.

Reports from the United States place the vaccine acceptance rate between 60% and 75% [8,12,14]. In Lebanon on the other hand, where the studies are fewer, the numbers are much more varied, ranging from a low of 21.4% to as high as 87% [15,16]. We found a high acceptance rate of 87.8%, which was similar in both groups. However, vaccine rejection was more prevalent among US participants (4.6% vs. 1.2%), with the Lebanese participants being more on the undecided side.

Misinformation regarding COVID-19 vaccines is rampant. In a national study in the United States, people who doubted the safety of the COVID-19 vaccine thought that it would be used to control people, to track their locations, or to change human DNA [17]. In our study, most respondents in both cohorts had similar misinformation, with only around 7–13% having correct knowledge about the vaccine (Table 2). Knowledge, or lack of it, is an important, albeit non-sufficient, factor that determines an individual’s perception of risk. This is particularly important when there is conflicting information available. Based on the ambiguity aversion theory, a person who receives conflicting information is unable to assign risk appropriately. In addition, a “known”, even if it carries a higher risk, is better perceived than an “unknown” which is safer [18]. This emphasizes the importance of combatting misinformation and providing people with venues through which they could acquire appropriate and accurate knowledge.

When asked about attitudes and perceptions about COVID-19 infection and vaccines, most study subjects believed in the benefit of routine vaccinations and in the effectiveness of vaccines in controlling transmissible diseases (Table 3). The COVID-19 vaccination concerns are particular to this vaccine and are not related to an anti-vaccine sentiment. The fact that a significant proportion of subjects believed that the COVID-19 vaccine development was rushed serves as a further confirmation that people are uniquely concerned about this vaccine. It is an incontrovertible fact that the COVID-19 vaccines were the fastest vaccines ever to be developed; some people may therefore assume that safety was overlooked in favor of efficacy [10]. In one study, vaccination acceptance was more likely if the amount of time spent in clinical trials increased [19]. Since there is considerable mistrust in the pharmaceutical industry [20,21], it becomes incumbent upon healthcare providers to explain the science to the general public and to assuage undue concerns about safety. Indeed, most respondents in our study stated that they would receive the COVID-19 vaccine if their healthcare provider recommended it. Another aspect that complicates the perception of the general public is the use of social media. We found that a large proportion of our study subjects (more in the US cohort than in the Lebanese cohort) obtained their health information from social media. In a recent narrative review, COVID-19 vaccine hesitancy correlated positively with reliance on social media as a source of information [22]. Given its powerful influence, it is therefore important to use social media as a means of disseminating reliable scientific information in lieu of its use to propagate conspiracy theories.

The independent determinants of vaccine hesitancy were more extensive in the US cohort, mainly because of the larger sample size, but all of the variables in the Lebanese regression model also featured in the US model and in the same order of importance (Table 4). The 3C model of vaccine hesitancy was first introduced in 2015 where Confidence, Complacency, and Convenience are used to classify the underpinnings of vaccine hesitancy [23]. Looking at our results, the variable having the strongest association with hesitancy (COVID-19 vaccine being more dangerous than the infection) falls within the Complacency category. Therein is a perception that the vaccine is not needed, i.e., that the risk posed by the infection does not justify the potential risk incurred by the vaccine. Another variable from the US regression model (COVID-19 infection does not worry me) is also in the same category. In the study by Kricorian et al., 20% of Americans questioning the safety of the vaccine thought that the pandemic was being exaggerated, and 14% thought that the vaccine was not needed [17]. In another report from the United States, the perception of high risk of acquiring the infection, high severity of the infection, and high effectiveness of the vaccine were all predictive of vaccine acceptance [12]. This is also consistent with data published from Lebanon [24,25]. The remaining important variables in our models (healthcare provider advice, rushed vaccine development, and laboratory origin of the virus) all relate to Confidence, which refers to the trust in the healthcare system and its agents. Numerous studies have established that mistrust in health systems is a powerful deterrent to vaccination [6,26,27]. In a Turkish study examining the psychological factors affecting COVID-19 vaccine hesitancy, participants who believed in conspiracy theories were less likely to accept vaccination [28]. In addition, a recent large survey from the United States reported that individuals who rejected the vaccine often attributed their decision to categorical beliefs against vaccination (termed “dissent”) or to cynicism towards the government or medical establishment (termed “distrust”) [29]. Interestingly, being a healthcare worker does not necessarily correlate with increased trust in the health system. Studies have revealed that healthcare workers who were not willing to be vaccinated believed in the rushed vaccine development [30] and questioned the reliability of the vaccine manufacturers as well as the novelty of the vaccines [31]. Despite an apparent lack of trust in the system, our study suggests that the participants have a high level of trust in their healthcare providers, seeing that a positive vaccine recommendation would increase the willingness to receive the vaccine, whereas a negative recommendation would have the opposite effect. This is consistent with the literature, where a positive and trusting relationship with healthcare providers was shown to boost vaccine acceptance [32,33]. As for Convenience, the third element of the 3C model which refers to difficulties in obtaining the vaccine, our survey did not address vaccine access issues, which prevents us from making any conclusions regarding their effect.

The main limitation of our study is that it used a convenience sample, indicating that it might not be totally representative of the American and Lebanese societal behavior. However, this is, to our knowledge, one of only a few studies that directly compared a population from a developed country to that from a developing country using the same instrument.

## 5. Conclusions

We found a relatively low vaccine hesitancy rate among US and Lebanese respondents. A number of determinants of hesitancy were identified, the two most important of which are the perception of high risk from the vaccine and negative vaccine recommendation by healthcare workers, whereas a positive recommendation by healthcare workers was associated with decreased hesitancy. Despite the obvious socio-behavioral differences between the US and Lebanese cohorts, the factors associated with hesitancy were similar for the most part in the two populations. The identified determinants could serve to inform vaccination programs in order to curb vaccination hesitancy.

## Figures and Tables

**Table 1 microorganisms-12-01200-t001:** Socio-demographic and health-related characteristics of the study population.

Variable	US Cohort (N = 6775)	Lebanese Cohort (N = 422)	Total (N = 7197)	*p*-Value
Age, years				<0.001
18–39	2358/6644 (35.5)	308/422 (73.0)	266/7066 (37.7)	
40–59	2080/6644 (31.3)	93/422 (22.0)	2173/7066 (30.8)	
≥60	2206/6644 (33.2)	21/422 (5.0)	2227/7066 (31.5)	
Gender				<0.001
Male	1442/6443 (21.7)	128/422 (30.3)	1570/7065 (22.2)	
Female	5055/6443 (76.1)	291/422 (69.0)	5346/7065 (75.7)	
Other	146/6443 (2.2)	3/422 (0.7)	149/7065 (2.1)	
Healthcare worker	1807/6443 (26.7)	124/422 (29.4)	1931/7065 (26.8)	0.22
Education				<0.001
Less than high school	21/6654 (0.3)	0/422 (0.0)	21/7076 (0.3)	
Completed high school	1113/6654 (16.7)	28/422 (6.6)	1141/7076 (16.1)	
Undergraduate degree	2660/6654 (40)	129/422 (30.6)	2789/7076 (39.4)	
Graduate degree	2226/6654 (33.5)	265/422 (62.8)	2491/7076 (35.2)	
Doctoral	634/6654 (9.5)	0/422 (0.0)	634/7076 (9.0)	
Alcohol consumption				<0.001
Never	2665/6638 (40.1)	24/422 (57.6)	2908/7060 (41.2)	
1–2 drinks/week	2322/6638 (35.0)	156/422 (37.0)	2478/7080 (35.1)	
3–5 drinks/week	947/6638 (14.3)	17/422 (4.0)	964/7060 (13.7)	
>5 drinks/week	704/6638 (10.6)	6/422 (1.4)	710/7060 (10.1)	
Smoking				<0.001
Never smoked	5215/6648 (93.5)	330/422 (78.2)	6545/7070 (92.6)	
<1/2 pack daily	252/6648 (3.8)	54/422 (12.8)	306/7070 (4.3)	
>1/2 pack daily	181/6648 (2.7)	38/422 (9)	219/7070 (3.1)	
Exercise				<0.001
Never	2543/6665 (38.2)	112/422 (26.5)	2655/7087 (37.5)	
≤3 days/week	2689/6665(40.3)	225/422 (53.3)	2914/7087 (41.1)	
>3 days/week	1433/6665 (21.5)	85/422 (20.1)	1518/7087 (21.4)	
Health status				<0.001
Healthy	2554/6775 (37.7)	323/422 (76.5)	2877/7197 (40.0)	
Preexisting condition	4221/6775 (62.3)	99/422 (23.5)	4320/7197 (60.0)	
Health perception				0.61
Poor	444/6627 (6.7)	24/422 (5.7)	468/7049 (6.6)	
Good	3262/6627 (49.2)	221/422 (52.4)	3483/7049 (49.4)	
Very good	2247/6627 (33.9)	137/422 (32.5)	2384/7049 (33.8)	
Excellent	674/6627 (10.2)	40/422 (9.5)	714/7049 (10.1)	
Previous personal diagnosis of COVID-19	539/6654 (8.1)	95/422 (22.5)	634/7076 (9.0)	<0.001
History of COVID-19 in family or friends	4259/6653 (64.0)	339/422 (80.3)	4528/7075 (65.0)	<0.001
How often do you take the influenza vaccine?				<0.001
Never	882/6649 (13.3)	209/422 (49.5)	1091/7071 (15.4)	
Yearly	5070/6649 (76.3)	108/422 (25.6)	5178/7071 (73.2)	
Once	697/6649 (10.5)	105/422 (24.9)	802/7071 (11.3)	
I am willing to take the COVID-19 vaccine				<0.001
Definitely yes	5846/6655 (87.8)	368/422 (87.2)	6214/7077 (87.8)	
Definitely no	305/6655 (4.6)	5/422 (1.2)	310/7077 (4.4)	
Undecided/likely	286/6655 (4.3)	33/422 (7.8)	319/7077 (4.5)	
Undecided/unlikely	218/6655 (3.3)	16/422 (3.8)	23/7077 (3.3)	

**Table 2 microorganisms-12-01200-t002:** Knowledge about COVID-19 infection and vaccination in the study population *.

Knowledge Question	US Cohort (N = 6775)	Lebanese Cohort (N = 422)	Total (N = 7197)	*p*-Value
*Questions where the expected answer is “True”*				
COVID-19 symptoms may include cough, diarrhea, and fever	6197/6775 (91.5)	4/18/422 (99.1)	6615/7197 (91.9)	<0.001
COVID-19 can present without fever	6231/6775 (92.0)	396/422 (93.6)	6626/7197 (92.1)	0.23
COVID-19 spreads through respiratory droplets produced when an infected person coughs, sneezes or talks	6585/6775 (97.2)	416/422 (98.6)	7001/7197 (97.3)	0.09
*Questions where the expected answer is “False”*				
Flu vaccine provides some protection against COVID-19 as well	6269/6775 (92.5)	401/422 (95.0)	6670/7197 (92.7)	0.06
Patients who recover from COVID-19 infection develop lifelong immunity	6652/6775 (98.2)	410/422 (97.2)	7062/7197 (98.1)	0.14
If I have allergies, I should not take the COVID-19 vaccine	6397/6775 (94.4)	380/422 (90.0)	6777/7197 (94.2)	<0.001
During the trials, serious adverse events and deaths occurred as a result of the vaccine	5487/6775 (81.0)	353/422 (83.6)	5840/7197 (81.1)	0.17
The COVID-19 vaccine can change human DNA	816/6775 (12.0)	161/422 (38.2)	977/7197 (13.6)	<0.001
The COVID-19 vaccine can sometimes lead to infertility	382/6775 (5.6)	149/422 (35.5)	531/7197 (7.4)	<0.001
The COVID-19 vaccine can be used to acquire my health data or track me	344/6775 (5.1)	172/422 (40.8)	516/7197 (7.2)	<0.001

* Numbers represent proportion and percentage of subjects who answered the question accurately.

**Table 3 microorganisms-12-01200-t003:** Attitudes and perceptions about COVID-19 infection and vaccination in the study population.

Attitude and Perception	US Cohort (N = 6775)	Lebanese Cohort (N = 422)	Total (N = 7197)	*p*-Value
The benefits from most routine vaccinations outweigh the risks or side effects	6372/6622 (96.2)	365/422 (86.5)	6737/7044 (95.8)	<0.001
Vaccination is an effective strategy to control the pandemic	6241/6610 (94.4)	414/422 (98.1)	6655/7032 (94.6)	0.001
The virus causing COVID-19 was created in a laboratory	1168/6515 (17.9)	153/422 (36.3)	1321/6937 (19.0)	<0.001
The COVID-19 vaccine development was rushed	2506/6635 (37.8)	217/422 (51.4)	2723/7057 (38.6)	<0.001
The COVID-19 vaccine is more dangerous than the COVID-19 infection	208/6598 (3.2)	22/422 (5.2)	230/7020 (3.3)	0.21
COVID-19 infection does not worry me	1889/6624 (28.5)	75/417 (18.0)	1964/7041 (27.9)	<0.001
My likelihood of becoming infected with COVID-19 is				<0.001
Very likely	442/6605 (6.7)	103/422 (24.4)	545/7072 (7.8)	
Likely	2112/6605 (32.0)	217/422 (51.4)	2329/7027 (33.1)	
Not likely	4051/6605 (61.3)	102/422 (24.4)	4153/7027 (59.1)	
How would you rate your knowledge of COVID-19?				<0.001
Excellent	777/6617 (11.7)	7/422 (1.7)	784/7039 (11.1)	
Very good	2637/6617 (39.9)	158/422 (37.4)	2795/7039 (39.7)	
Good	3975/6617 (45.0)	170/422 (40.3)	3145/7039 (44.7)	
Poor	228/6617 (3.4)	87/422 (20.6)	315/7039 (14.5)	
I would get the COVID-19 vaccine if my healthcare provider recommends it	5969/6588 (90.6)	349/419 (83.3)	6318/7007 (90.2)	<0.001
My healthcare provider advised me not to take the COVID-19 vaccine	87/6216 (1.4)	12/422 (2.8)	99/6638 (1.5)	0.18
Main sources of health information				
Healthcare provider	5272/6775 (77.8)	275/422 (65.2)	5547/7197 (77.1)	<0.001
Family	4562/6775 (67.3)	123/422 (29.1)	4685/7197 (65.1)	<0.001
General media	4426/6775 (65.1)	137/422 (32.5)	4563/7197 (63.4)	<0.001
Social media	4186/6775 (61.8)	147/422 (34.8)	4333/7197 (60.2)	<0.001

**Table 4 microorganisms-12-01200-t004:** Multivariable regression analysis of potential determinants of vaccine hesitancy in the US and Lebanese cohorts *.

Variable	US Cohort (N = 6775)		Lebanese Cohort (N = 422)	
	aOR (95% CI)	*p*-Value	aOR (95% CI)	*p*-Value
The COVID-19 vaccine is more dangerous than the COVID-19 infection	8.7 (3.1–24.3)	<0.001	9.4 (3.3–26.7)	<0.001
My healthcare provider advised me not to take the COVID-19 vaccine	6.5 (2.9–14.2)	<0.001	5.4 (1.0–28.4)	0.04
The benefits from most routine vaccinations outweigh the risks or side effects	5.4 (2.7–10.8)	<0.001		
The COVID-19 vaccine development was rushed	3.2 (2.4–4.2)	<0.001	3.7 (1.7–8.2)	<0.001
The virus causing COVID-19 was created in a laboratory	2.9 (2.2–4.0)	<0.001		
Healthcare worker	2.2 (1.6–3.2)	<0.001	3.3 (1.2–9.0)	0.02
COVID-19 infection does not worry me	2.0 (1.5–2.7)	<0.001		
Previous personal COVID diagnosis	1.6 (1.1–2.5)	0.02		
Smoking	1.6 (1.2–2.2)	0.001		
How would you rate your knowledge of COVID-19	1.4 (1.1–1.7)	0.001		
Knowledge score	1.3 (1.1–1.5)	0.002	0.7 (0.6–0.9)	0.02
Education	0.8 (0.7–0.9)	0.004	0.6 (0.3–0.9)	0.02
How often do you take the influenza vaccine?	0.6 (0.5–0.8)	<0.001		
Age	0.6 (0.5–0.7)	<0.001	0.4 (0.2–0.8)	0.01
Vaccination is an effective strategy to control the pandemic	0.3 (0.2–0.5)	<0.001		
I would get the COVID-19 vaccine if my healthcare provider recommends it	0.02 (0.01–0.03)	<0.001	0.4 (0.2–0.9)	0.03

* Vaccine hesitancy was defined as any answer other than “definitely yes” to “I am willing to take the COVID-19 vaccine” and includes the answers “definitely no”, “undecided/unlikely”, and “undecided/likely”. aOR = adjusted odds ratio; CI = confidence interval.

## Data Availability

The raw data supporting the conclusions of this article will be made available by the authors upon request.

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
