# Peer review of "Determinants of COVID-19 Vaccine Hesitancy: A Cross-Sectional Study in Three Communities in the United States and Lebanon"

_microorganisms, 2024, doi:10.3390/microorganisms12061200_

Round 1

Reviewer 1 Report

Comments and Suggestions for Authors

I support the value of the work: cross sectional survey in two countries allowing the identification of the major determinants of vaccination uptake and vaccination refusal, as well as the relevance of the moment: peak months of the COVID-19 pandemic (April 2021 – June 2021), when a gradually rising proportion of the population in both countries had access to COVID-19 vaccines, when a significant fraction was reluctant to receive the vaccine. Estimating vaccine hesitancy and its foundations is definitively important in informing future vaccination campaigns.

In the U.S., vaccine hesitancy was greater for Black and Hispanic participants and those reporting more than one or other race (https://www.nature.com/articles/s41467-022-28200-3.pdf). As a reviewer I consider that the race need to be addressed among the variables of the study, discussed or justified why the race was not considered among the variables. In addition, race should be considered in the table of the Socio-demographic characteristics of the study population for the analysis or alternatively well justified why this is out from the table.

These figures can change rapidly because new information becomes available and as vaccination campaigns progress, it would be of value to provide recommendations to public health officials to work on addressing vaccine hesitancy. Author’s perspective on the progress of this problem on time would also be of value, and of course considering the problem of minorities and race.

Author Response

Thank you see attached file

Reviewer 2 Report

Comments and Suggestions for Authors

The aim of this study was to compare the determinants of COVID-19 vaccination  in the United States and Lebanon. The study was quite interesting for the common reader of this journal.

Few comments needed to be explored in the manuscript

1. Why the authors compare US with LEBANON, possible reason should be mention in the introduction section

2. The authors should mention about the ethical guidelines or any approval before conducting this study, I did not get any specific number for this

3. The first paragraph of the discussion needed to be improved and also they can insert the table number at appropriate places in the discussion section

4. All references should be modified based on the MDPI guidelines.

Author Response

Thank you for your review. See attached document. 

Round 2

Reviewer 1 Report

Comments and Suggestions for Authors

I accept authors' response. The manuscript is acceptable for publication.